# The Pathophysiological Significance of “Mitochondrial Ejection” from Cells

**DOI:** 10.3390/biom12121770

**Published:** 2022-11-28

**Authors:** Qintao Fan, Yasuhiro Maejima, Lai Wei, Shun Nakagama, Yuka Shiheido-Watanabe, Tetsuo Sasano

**Affiliations:** Department of Cardiovascular Medicine, Graduate School of Medical and Dental Sciences, Tokyo Medical and Dental University, Tokyo 113-8510, Japan

**Keywords:** mitochondria, exopher, tunneling nanotubes, extracellular vesicles, heart, mitochondrial transfer, mitochondrial quality control

## Abstract

Mitochondria have beneficial effects on cells by producing ATP and contributing to various biosynthetic procedures. On the other hand, dysfunctional mitochondria have detrimental effects on cells by inducing cellular damage, inflammation, and causing apoptosis in response to various stimuli. Therefore, a series of mitochondrial quality control pathways are required for the physiological state of cells to be maintained. Recent research has provided solid evidence to support that mitochondria are ejected from cells for transcellular degradation or transferred to other cells as metabolic support or regulatory messengers. In this review, we summarize the current understanding of the regulation of mitochondrial transmigration across the plasma membranes and discuss the functional significance of this unexpected phenomenon, with an additional focus on the impact on the pathogenesis of cardiovascular diseases. We also provide some perspective concerning the unrevealed mechanisms underlying mitochondrial ejection as well as existing problems and challenges concerning the therapeutic application of mitochondrial ejection.

## 1. Introduction

Mitochondria are double membrane-bound α-proteobacterium-derived organelles that contain extrachromosomal deoxyribonucleic acid (DNA), termed mitochondrial DNA (mtDNA) [1]. Most of the genes that encode mitochondrial proteins have been transferred to the genome in the nucleus of host cells, with only 16 kilobases of circular DNA left in the mitochondria [2]. Mitochondria play a vital role by producing adenosine triphosphate (ATP), an essential energy source for all organisms. Concurrently with the adenosine triphosphate (ATP) generation, mitochondria are closely associated with other physiological metabolic pathways, including glycolysis, calcium homeostasis, steroid synthesis, and cell cycle regulation. On the other hand, mitochondria also participate in a series of pathological processes [3]. Reactive oxygen species (ROS), which cause damage to proteins, nucleic acids, and lipids in the host cells, are produced as byproducts of the OXPHOS process [4]. As a result of mitochondrial damage, mtDNA can be released, thereby playing a detrimental role by triggering various proinflammatory signaling pathways [5,6]. A series of mitochondrial quality control systems, namely the ubiquitin-proteasome system (UPS), mitochondrial dynamics, and mitochondria-selective autophagy (mitophagy) remove damaged mitochondrial components [7,8,9,10]. As cardiomyocytes (CMs) are post-mitotic cells that require long-term survival, it is essential to maintain tight control over the quality of mitochondria. Thus, impairment of the mitochondrial quality control systems provokes critical cardiovascular diseases, including heart failure [11,12,13].

Recent investigations have revealed that unexpected mechanisms are involved in maintaining mitochondrial quality in cells by transporting mitochondria between the inside and outside of cells and across plasma membranes. For example, it has been shown that nerve cells, CMs, and adipocytes are able to eject damaged mitochondria from their cytosol [14,15,16]. Macrophages have dual roles in the uptake and elimination of mitochondria. Stem cells are able to donate healthy mitochondria that can enhance the aerobic respiration capacity of recipient cells [17]. In addition, mitocytosis, a special mitochondrial quality control process through which damaged mitochondria are transported into migrasomes for the disposal of them from migrating cells, has recently been reported [18]. In this review, we summarize the current understanding of the regulation of mitochondrial transmigration across plasma membranes and discuss the functional significance of this unexpected phenomenon, with a focus on the impact on the pathogenesis of cardiovascular diseases.

## 2. Disposal and Elimination of Damaged Mitochondria

In 2014, Davis et al. detected transcellular mitochondrial degradation from retinal ganglion cell axons to astrocytes under baseline conditions in vivo [19]. Using three-dimensional electron microscopy (EM), they identified axonal evulsions that contained mitochondria surrounded by astrocytes, and transmission EM revealed the presence of both intact mitochondrial structures and irregular mitochondria-like membranous bodies. Mitochondria shed into evulsions were taken and eliminated by adjacent astrocytes. These results implied that a similar process occurs in the murine cerebral cortex. A recent study demonstrated that damaged mitochondria were released from neurons to culture media [15]. The number of depolarized mitochondria released from the cytosol to extracellular spaces was markedly increased by treatment with rotenone or carbonyl cyanide m-chlorophenylhydrazone (CCCP), the inducers of mitochondrial stress, in PC12 cells [15]. The overexpression of Parkin (PRKN), a mitophagy regulator that is associated with hereditary Parkinson’s disease, or PRKN-independent regulators reduced the release of mitochondria [15]. On the other hand, the number of ejected mitochondria was markedly increased in autophagy-deficient cells [15]. Additionally, it has been shown that the number of mitochondria released into media was enhanced by acidotic conditions (the cells were incubated with pH6.5 solution) or oxidant-induced stress in mouse primary neurons [20]. Falchi et al. found that ejected mitochondria from human-cultured fetal astrocytes in large vesicles (1–8 μm) were eliminated by phagocytic cells with undetermined mechanisms [21].

According to a report by Melentijevic et al., the ejection of injured mitochondria occurred in a *Caenorhabditis elegans* (*C. elegans*) model [22]. The neurons of *C. elegans* eject “exophers”, membranous vesicles of 2–15 μm in diameter which contain mitochondria and other organelles or proteins. The production of exophers is provoked by various stresses, the suppression of autophagy or UPS activities, and mitochondrial damage. Recently, Nicolás-Ávila et al. revealed a transcellular mitochondrial degradation pathway between CMs and macrophages in healthy myocardium [14]. They found that CMs packed damaged mitochondria in LC3^+^ exophers for ejection to extracellular spaces in baseline, and isoproterenol-induced cardiac stress promoted the ejection of exophers (Figure 1A). Released exophers were phagocytized and degraded by cardiac-resident macrophages (cMacs). Such a phenomenon is termed “Heterophagy” [23]. Both the stimulation and suppression of autophagy could induce a parallel trend in the release of exopher, suggesting that the exopher-mediated disposal system of mitochondria would be controlled by autophagy-associated regulators [23] (Figure 1A). In accordance with apoptotic cells, phosphatidyl serine, an “eat-me” signal that is recognized by MER proto-oncogene tyrosine kinase (MERTK)—a type of tyrosine kinase that is highly expressed in cMacs—was found to be expressed on the surface of exophers. MERTK deletion caused the removal of exophers to become stuck, suggesting that MERTK would be a receptor of exophers in cMacs [23]. In addition, the ablation of cMacs caused the accumulation of exophers, which resulted in the provocation of cardiac inflammation, suggesting that the release of exophers plays a critical role in maintaining a healthy condition in the heart [23].

Rosina et al. revealed that the transcellular degradation of damaged mitochondria between brown adipocytes and brown adipose tissue (BAT)-resident macrophages was mediated through extracellular vesicles (EVs) [16]. Under cold stress, brown adipocytes eject oxidatively-injured mitochondria via EVs, which diminish the mitochondrial function after their re-uptake by other brown adipocytes. The clearance of these EVs is executed by BAT-resident macrophages. The in vivo depletion of macrophages resulted in the accumulation of mitochondria-containing EVs and, subsequently, in impaired thermogenesis in BAT, suggesting the importance of this process in maintaining thermogenic homeostasis in BATs [16]. This study differs from the previous findings of Crewe et al. since EVs containing functional but damaged mitochondria, which were produced by stressed adipocytes, were found to enter isolated CMs, thereby promoting the production of ROS while also having a compensatory antioxidant effect in the heart [24].

Mitocytosis is a migrasome-mediated mitochondrial disposal process that is regarded as a crucial mitochondrial quality control system in migrating cells [18]. When cells start migrating, tubular structures termed “retraction fibers” grow from the end of the tail. In the next step of cell migration, cellular components are transferred, thereby accumulating in retraction fibers. Then, accumulated cellular components form large vesicular organelles called “migrasomes”. Finally, when the cells move away, migrasomes are left behind or taken up by adjacent cells [25] (Figure 1B). Jiao et al. found the existence of mitochondria with clearly visible cristae-like structures in migrasomes induced by CCCP-treated L929 cells [18]. Time-lapse imaging enabled them to observe that tubular-form mitochondria were transported outward with extension in a highly dynamic manner [25]. Part of the mitochondria that moved to the peripheral area of the cells appeared to be separated from the mitochondrial network and was captured in the plasma membrane, and was thereby transferred into migrasomes wherein the host cells migrate away. Intriguingly, damaged mitochondria—as evidenced by decreased membrane potential with a high ROS production—are selectively transported to the peripheral area of cells, and are thereby released into migrasomes [25]. Damaged mitochondria have a weakened ability to rescue dynein, the inward motor, while their ability to bind kinesin family member 5B (KIF5B) significantly increased. Therefore, in cooperation with mitochondrial fission regulator dynamin-related protein 1 (Drp1) and myosin XIX (Myo19), damaged mitochondria are transferred to the peripheral area of cells and then are disposed of by migrasomes [25] (Figure 1B). Thus, the mitocytosis-mediated mitochondrial disposal process plays a critical role in the maintenance of mitochondrial quality and intracellular homeostasis.

## 3. Role of Released Mitochondria as a “Savior” of Target Cells

The intercellular transport of mitochondria was first reported by Koyanagi et al. in 2005 [26]. A subsequent study demonstrated the importance of mitochondrial transfer in vitro which improved the aerobic respiration of mtRNA-depleted recipient cells [27]. Accumulating lines of evidence suggest that mesenchymal stem cells (MSCs) play a role in the protection of various other cells by providing their intact mitochondria to other damaged cells via intercellular mitochondrial transportation. For instance, MSCs have the ability to rescue CMs from ischemia-induced damage via mitochondrial transportation, which is mediated through tunneling nanotubes [28,29]. Mitochondria—but not cytoplasm—could be exclusively transferred toward CMs [30]. O’Brien et al. demonstrated that MSCs were able to transport mitochondria through large EVs to doxorubicin-treated patient-specific induced pluripotent stem cell-derived CMs (iCMs), thereby improving contractility and the mitochondrial function of acceptor cells [31].

It has been shown that MSCs alleviated airway epithelial cells from lipopolysaccharide (LPS)-induced acute lung injury [32] and asthma inflammation [33] by providing intact mitochondria to damaged cells. Similarly, Li et al. revealed that transplanted induced pluripotent stem cells (iPSC)-MSCs rescued airway epithelial cells from cigarette smoke-induced damage via the transference of mitochondria [34]. The salutary roles of mitochondrial transfer from MSCs to neurons have been detected. Transferred mitochondria from MSCs were able to repair the respiration of recipient neurons, thereby reducing the ischemic damage [35] or oxidant injury [36] of damaged neurons. Further research demonstrated the ability of MSCs to transfer mitochondria to astrocytes [37] and neural stem cells [38] for exerting a protective effect on the recipient cells.

Recent investigations revealed that the intercellular transport of mitochondria could also be seen between mature cells. Astrocytes transport mitochondria to neurons to protect the neuronal function in response to transient focal cerebral ischemia in mice [39] and cisplatin-treated cultured neurons [40]. Mitochondria are able to vigorously transfer between neurons and astrocytes [41] or brain endothelial cells and neurons [42]. Similarly, Shen et al. reported that mitochondrial transfer between neonatal rat CMs and cardiac myofibroblasts (MFs) protected CMs from damage-induced apoptosis [43,44]. A recent report demonstrated that human iCMs are able to produce mitochondria-containing EVs. The intramyocardial injection of mitochondria-containing EVs isolated from the culture media of iCMs relieved cardiac dysfunction in the murine post-infarction heart failure model, suggesting that the transplantation of mitochondria-containing EVs to the CMs could be a potential novel therapeutic strategy for post-infarction heart failure [45]. Mitochondrial transport between matured cells also occurs in the kidneys [46] and osseous tissue [47], where it restores the metabolic function of recipient cells.

The intercellular transport of mitochondria is also involved in the genesis, survival, metastatic transformation, and drug resistance of malignant tumors [48]. Thus far, mitochondrial transportation has been observed in breast cancer, lung cancer, melanoma, and leukemia. It has been shown that malignant cells receive mitochondria from non-malignant cells to restore their respiratory function [49,50] and/or acquire chemoresistance [49,51,52]. Recently, Saha et al. demonstrated an intriguing phenomenon wherein cancer cells “hijack” mitochondria from immune cells by sucking through nanotubes, thereby improving their own metabolic activity and causing immune cell death [53].

## 4. Released Mitochondria-Mediated Regulation of the Immune System

It is widely recognized that mitochondria exert an immunomodulatory function through various mechanisms. The metabolic activities and metabolites of mitochondria are related to the regulation of immune activities of various immune cells. Similarly, mtDNA is closely associated with immunoreaction as a damage-associated molecular pattern (DAMP). Recently, immunoregulation mediated by “intercellular mitochondrial transfer” has received attention. Boudreau et al. substantiated the existence of functional mitochondria in EVs released by platelets, which was found in platelet concentrates used for transfusion [54]. They also found that the hydrolysis of such extracellular mitochondria leads to the release of phospholipase A2 IIA (sPLA2-IIA), which results in the promotion of inflammation. Two-photon microscopy revealed that extracellular mitochondria triggered the adhesion of neutrophils, suggesting that extracellular mitochondria play an important role in the initiation of inflammatory responses. Puhm et al. demonstrated that monocytes could release mitochondria-contained EVs in response to LPS-mediated activation [55]. They also revealed that any mitochondria directly isolated from LPS-activated monocytes, mitochondria-containing EVs isolated from monocytes, and circulating EVs isolated from volunteers receiving low-dose LPS injections could provoke type I *Interferon* (IFN) and *tumor necrosis factor* (TNF) responses in endothelial cells. On the other hand, the loss of mitochondrial respiratory capacity, the presence of pyruvate, and MitoTEMPO, a mitochondrial reactive oxygen species scavenger, dramatically reduced the inflammatory response. Hough et al. reported that mitochondria-containing EVs could be seen in the bronchoalveolar lavage fluid of both healthy volunteers and asthmatics [56]. Although it is possible that those mitochondria-containing EVs were released from airway myeloid-derived regulatory cells and internalized by recipient T cells, their functions remain unclear.

A growing body of evidence suggests that mitochondrial transport by MSCs plays an important role in the regulation of immune responses. In an animal model of acute respiratory distress syndrome (ARDS), MSCs assisted the removal of bacteria by transferring mitochondria to macrophages via tunneling nanotubes (TNTs) [57] or mitochondria-containing EVs [58]. Mitochondrial transport could induce the differentiation of macrophages to the anti-inflammatory M2 phenotype with enhanced phagocytosis ability, thereby reducing the inflammatory response [58,59]. In the case of primary CD4^+^CCR6^+^CD45RO^+^ T helper 17 (Th17) cells cocultured with MSCs, transported mitochondria from MSCs caused interconversion of Th17 cells into T regulatory cells, which resulted in the significant impairment of IL-17 production from Th17 cells [60]. A subsequent investigation demonstrated that MSC-derived mitochondria enhanced the expression of FOXP3, IL2RA, CTLA4, and TGFβ1 in CD4^+^ T cells, thereby promoting the activation of T-cells or differentiation into regulatory T cells. Indeed, the transplantation of MSC-derived mitochondria-treated CD4^+^ T cells significantly reduced the organ damage caused by inflammation and improved the survival rate in murine *graft-versus-host disease* models [61]. In addition, it has been shown that MSC-derived mitochondria play a suppressive role in the activity of autophagy in T cells, thereby inhibiting the apoptosis of T cells caused by high autophagy activity [62].

## 5. Mechanisms of Mitochondrial Release/Transfer from Cells

Although numerous studies have revealed several routes of mitochondrial release/transfer, including TNTs and MVs, the specific molecular mechanisms remain to be elucidated. In this article, we have summarized the current understanding of the molecular mechanisms involved in the mitochondrial release and/or transfer from cells (Table 1).

TNTs, a unique intercellular membrane structure, were first reported in rat pheochromocytoma PC12 cells [70]. TNTs have a diameter of 50–200 nm, which allows the transport of cell membranes, plasma, and organelles, such as mitochondria, between cells through intercellular connected tubes (Figure 2A). A variety of stimuli that cause mitochondrial damage can induce the formation of TNTs. Mechanistically, p53 is considered to be an important upstream regulator in the process of TNT formation [71]. Activated p53 triggers the epidermal growth factor receptor activation, followed by the upregulation of the downstream Akt-phosphatidylinositol-3 kinase-mTOR pathway, leading to the high expression of M-sec (TNFαIP2). M-sec is one of the key regulators in the formation of TNTs [72], which is responsible for mediating the assembly of exocyst complexes by interacting with RalA to induce F-actin polymerization. CD38, a transmembrane glycoprotein known as cyclic ADP ribose hydrolase, is expected to be a key regulator of TNT-mediated mitochondrial transfer. In the co-culture model of primary multiple myeloma cells and BMSC, TNT-mediated mitochondrial transfer to multiple myeloma cells is inhibited by the shRNA-mediated knockdown of CD38 [49]. Wang et al. demonstrated that enhanced CD38 expression caused by the activation of extracellular regulated protein kinases 1/2 (ERK1/2) promoted mitochondrial transfer from astrocytes [73]. The intercellular concentration gradient of the S100 calcium-binding protein A4 (S100A4) is known to be closely associated with the process of TNT elongation. Namely, a previous report demonstrated that TNTs expand from the donor cells with lower concentrations of S100A4 to receptor cells with higher concentrations [74]. In addition, Cdc42 has been suggested to be involved in the prolongation of TNTs [72]. Gap junction protein connexin 43 (CX43) mediates mitochondrial transfer between bone marrow-derived stem cells and LPS-damaged alveolar epithelial cells [32], possibly through the promotion of TNT formation [33]. On the other hand, He et al. demonstrated that CX43 is absent in the TNTs between neonatal rat CMs and cardiac fibroblasts (CFs) [43], suggesting that further studies are needed to determine the specific role of CX43 in the formation of TNTs. Recent studies have revealed that Miro1, a mitochondrial outer membrane protein, plays an important role in regulating the intercellular movement of mitochondria along TNTs [36,37,68,69]. As Miro1 is a type of calcium-sensitive Rho-GTPase [75], it can be bound to TRAK1/2, which results in the mediation of the movement of mitochondria by recruiting a series of kinesins. In addition, Miro1 interacts with mitofusins (Mfn1 and Mfn2), thereby facilitating the transport of mitochondria along TNTs [76]. Shen et al. demonstrated that mitochondrial transport along TNTs is mediated by KIF5B, a member of the kinesin superfamily, in the co-culture system of neonatal rat CMs and CFs [44].

EV is another important transporter of mitochondria (Figure 2B). EVs are membranous structures derived from the endosomal system or shed from the plasma membrane, including exosomes and microvesicles, which differ in size and origin [77]. Previous studies have shown that cells can secrete EVs with mitochondrial components. Although the molecular mechanism needs to be further elucidated, a potential link between the mitochondria-containing EV and the mitochondrial-derived vesicle (MDV), a unique vesicle capable of transporting mitochondria to other organelles has been revealed [16,24,78]. MDVs are generated through the selective packing of mitochondrial content and transported to lysosomes to be degraded early on in mitochondrial damage, causing them to be considered an early response of mitochondrial quality control [79]. A recent study demonstrated that the packaging of mitochondrial proteins into EVs requires the expression of Snx9 and OPA1, while PRKN facilitates the delivery of MDVs containing damaged mitochondrial components to lysosomes rather than EVs, thus preventing the release of mitochondrial DAMPs [78]. Consistently, the inhibition of lysosomal activity promotes the release of mitochondria containing EVs in adipocytes [16,24]. However, according to previous studies, oxidatively damaged mitochondrial proteins are enriched in both MDVs and EVs isolated from stressed adipocytes, suggesting some dysfunction in the selective packaging process.

In addition to participating in TNT-mediated mitochondrial transfer, CD38 is involved in the extracellular release of mitochondrial particles by astrocytes [39]. As mentioned above, astrocytes promote the survival of neurons by producing mitochondrial EVs after a stroke, while the inhibition of CD38 was shown to significantly decrease the number of mitochondria containing EVs, suggesting that a CD38-dependent pathway is required for the ejection of these EVs.

Mitochondria can be ejected from cells or transferred to other cells through various other pathways. Activated platelets release functional mitochondria through the actin-dependent pathway, which is independent of the microtubule-dependent pathway. This process is considered to be a damage-associated mechanism [54]. LPS-treated monocytic cells extrude naked mitochondria with uncertain molecular mechanisms [55].

## 6. Concluding Remarks

In summary, we reviewed the functional significance of mitochondrial transmigration across the plasma membranes in various cells. We also summarized the trends in investigations of such phenomena as novel therapeutic targets of various diseases, including heart failure. In recent years, mitochondrial supplemental therapy—based on mitochondrial transfer—has been proven to improve the function of various damaged tissues and organs. Previous studies have verified the potential efficacy of MSC transplantation-mediated mitochondrial supplemental therapy in animal experiments [69,80]. With the emergence of new technologies, the focus of mitochondrial transfer therapeutics is gradually shifting from cell-dependent therapy to cell-free therapy. Indeed, EV-based mitochondrial transplantation has been shown to be capable of significantly improving the function of ischemic myocardium after treatment, and it is favored due to its stability in stressed environments [45]. However, there are several critical limitations to these therapeutic strategies. For example, considering that the mechanism of selectivity of mitochondrial transfer to recipient cells remains unclear, the safety issues caused by undesired transfer and its potential effects (unwanted redifferentiation, excess mitochondria, ATP in recipient cells, etc.) need to be further elucidated. In addition, although studies on mitochondrial transfer have been extensively conducted in the fields of immunology, cancer biology, and neuroscience, very few studies on mitochondrial transfer have been conducted in the field of cardiovascular science. As mitochondrial transfer therapy is expected to have great therapeutic potential in cardiovascular diseases, more attention is needed.

Increasing lines of evidence suggest that the ejection of damaged mitochondria plays an important role in maintaining the quality of intracellular mitochondria. However, our understanding of the molecular mechanisms involved in the regulation of such a process remains limited. Recent studies have shown that MDVs are involved in the extracellular ejection of damaged mitochondria via EVs [16,24]. However, the mechanism by which these EVs are transported outward and finally released is still poorly understood. For instance, the mechanism underlying the release of exophers from cells is completely unknown. The process through which damaged mitochondria are ejected remains unclear, but possible mechanisms include selective packing into MDV-like vesicles or autophagosomes before ejection into extracellular spaces or direct transportation via a mitocytosis-like process. Furthermore, we still know very little about how extracellular emissions and the intracellular mitochondrial quality control system share roles in the removal of damaged mitochondria. Cardiomyocytes rely heavily on mitochondrial quality control mechanisms to maintain their survival and cell function. Due to their huge energy demands and massive amounts of mitochondria, additional mechanics, such as the ejection of exophers, may be required to cope with stress and damage. Indeed, 60% of total mitochondria in healthy myocardium are ejected through exophers [14], a percentage that increases even further in response to stress, suggesting that this mechanism may be more adaptable than intracellular mitochondrial quality control mechanisms, including mitophagy, and act as a unique countermeasure to injury in cardiomyocytes (Figure 3). In addition, the expression of LC3 on the surface of exophers, together with the effect of autophagy regulation on the number of exophers, suggests that this mechanism is associated with autophagy machinery. The solution to these unsolved questions will help us to further understand the association between the release of damaged mitochondria and the pathogenesis of cardiovascular diseases, and ultimately promote the application of mitochondrial transplantation therapy for these conditions.

## Figures and Tables

**Figure 1 biomolecules-12-01770-f001:**
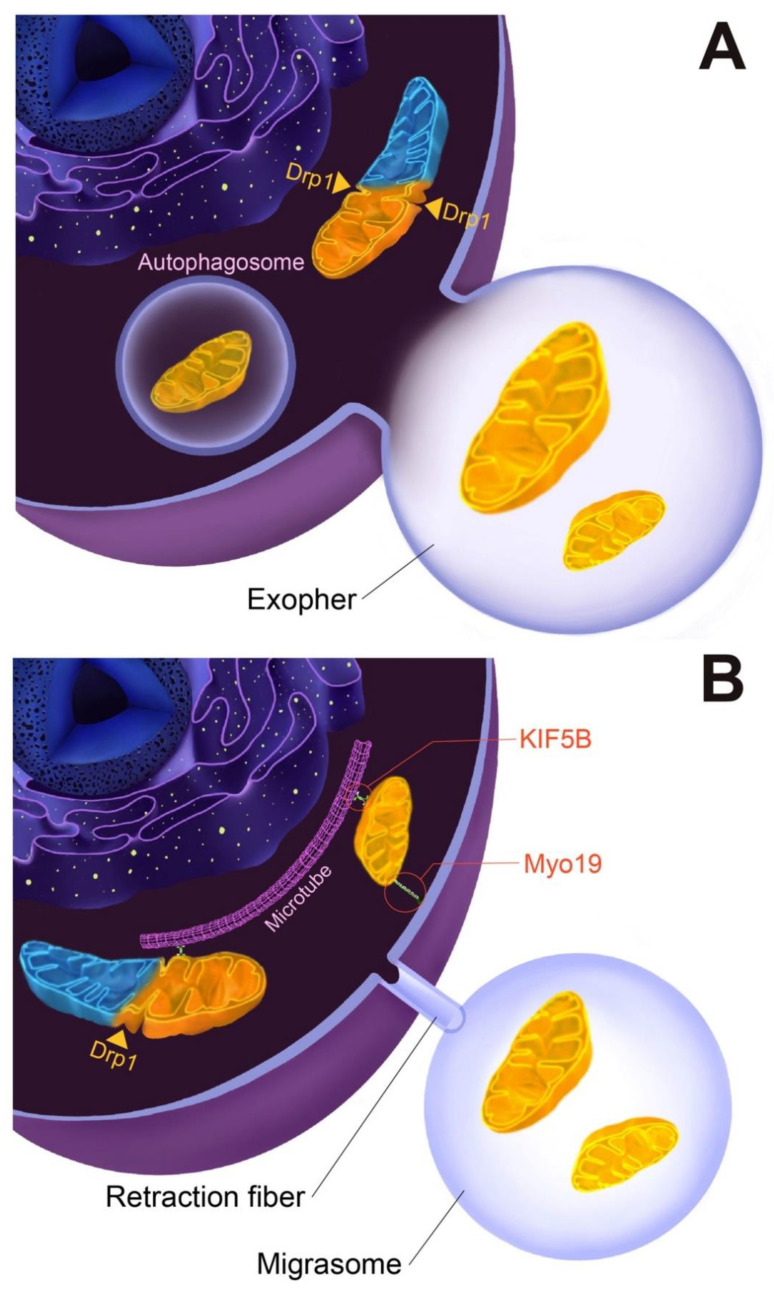
Schematic representations of mitochondrial disposal processes. (**A**) Exopher-mediated mitochondrial disposal process. (**B**) Mitocytosis, a migrasome-mediated mitochondrial disposal process. See text for details and definitions of abbreviations. Drp1: dynamin-related protein 1, KIF5B: kinesin family member 5B, Myo19: myosin XIX.

**Figure 2 biomolecules-12-01770-f002:**
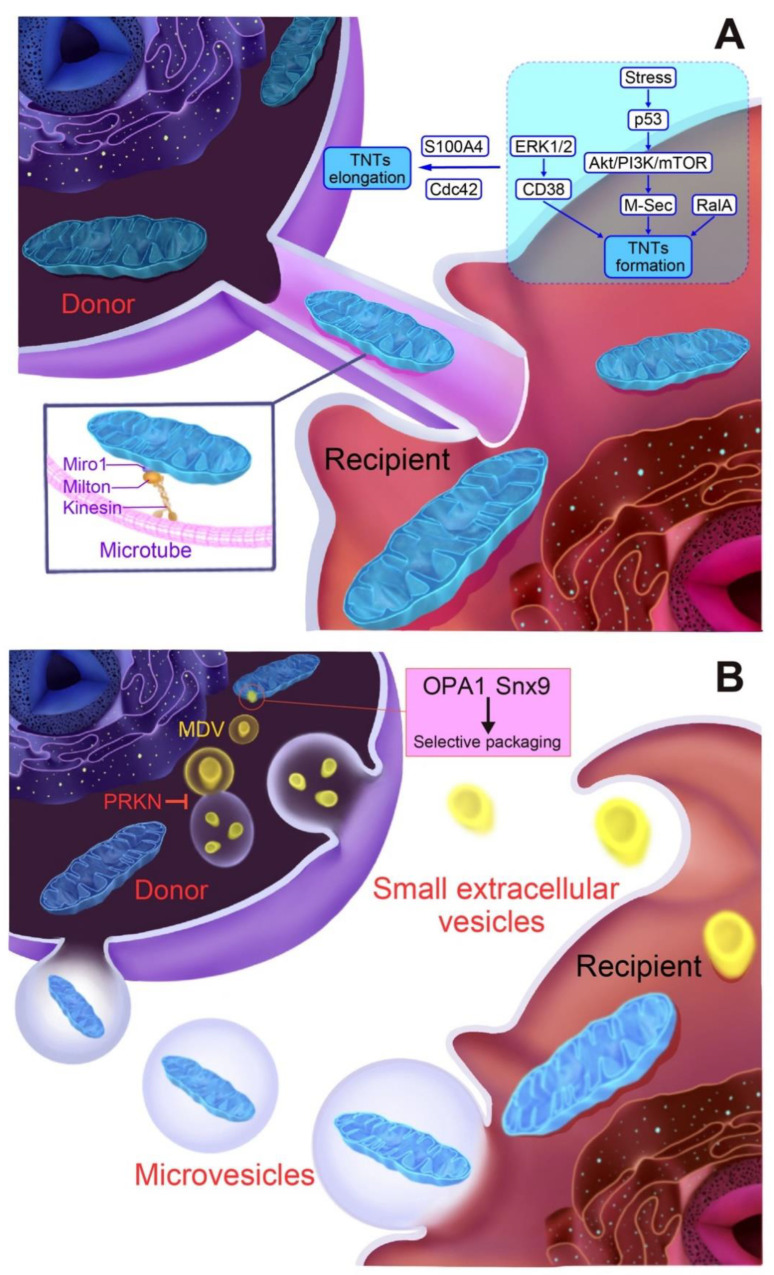
Schematic representations of mechanisms underlying mitochondrial release or transfer from cells. (**A**) Tunneling nanotube (TNT)-mediated intercellular transfer of mitochondria. (**B**) Extracellular vesicle (EV)-mediated mitochondrial release/transfer from cells. See text for details and definitions of abbreviations. CD38: cluster of differentiation 38, *Cdc42:* Cell division control protein 42, ERK1/2: extracellular regulated protein kinases 1/2, MDV: mitochondrial-derived vesicle, OPA1: Optic atrophy 1, PRKN: Parkin, *RalA*: Ras-related protein *A*. *S100A4*: S100 calcium binding protein A4, Snx9: Sorting nexin-9.

**Figure 3 biomolecules-12-01770-f003:**
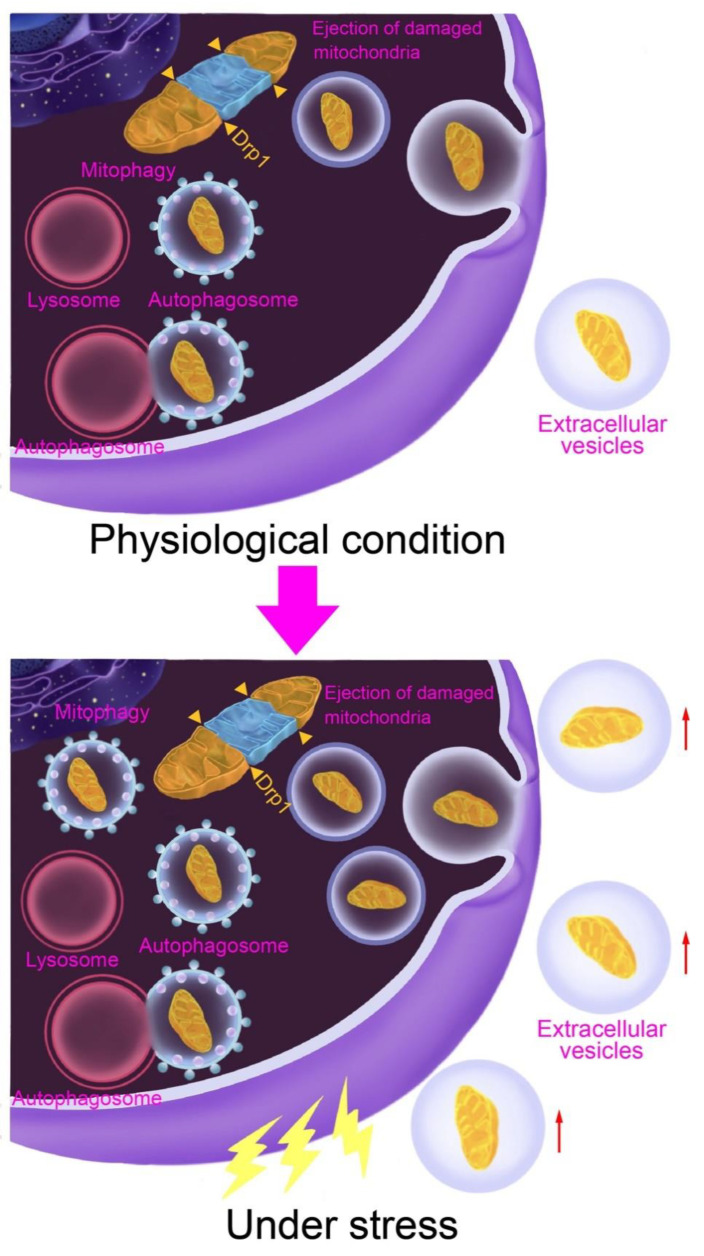
Schematic hypothetical model of the mitochondrial quality control system in cardiomyocytes. Under physiological conditions, both the intracellular mitochondrial quality control systems, such as mitophagy, and extracellular mitochondrial ejection equivalently contribute to the removal of damaged mitochondria. However, the demand for extracellular clearance of damaged mitochondria is markedly increased under stress conditions in cardiomyocytes. Explanation red arrow: increasing.

**Table 1 biomolecules-12-01770-t001:** Summary of the reports regarding mitochondrial release and/or transfer from various cells.

Donor	Acceptor	Stimulation	Components	Outcome	Routes	Ref.
Adipocytes	Cardiomyocytes	Mitochondrial stress	Damaged mitochondria	Burst of ROS; protect the heart through hormesis	EVs	[24]
hMSCs	A549 cell line (NSCLC-derived)		Mitochondria and mtDNA	Rescued aerobic respiration		[27]
MSCs	Cardiomyoblasts	Ischemia	Mitochondria	Reduced cell death		[28]
BMMSCs	H9c2 cell line	Ischemia/reperfusion	Mitochondria	Reduced apoptosis	TNTs	[29]
MSCs	Cardiomyocytes		Mitochondria			[30]
MSCs	ICMs	Doxorubicin administration	Mitochondria	Improved ICM contractility and mitochondrial biogenesis	EVs	[31]
BMSCs	Alveolar epithelia	Acute lung injury (ALI) caused by lipopolysaccharide (LPS)	Mitochondria	Reduced acute lung injury, increased alveolar ATP concentrations	Gap junctions and MVs	[32]
iPSC-MSCs	Epithelial cells		Mitochondria	Improved mitochondrial function, reduced inflammation	TNTs	[33]
iPSC-MSCs	Epithelial cells	Cigarette smoke	Mitochondria	Alleviates alveolar fibrosis	TNTs	[34]
MMSCs	Neuronal cells		Mitochondria	Alleviated stroke-induced damage		[35]
MSCs	Mouse neurons	Hydrogen peroxide exposure	Mitochondria	Increased neuronal survival and improved metabolism		[36]
MMSCs	Neurons	Oxygen-glucose deprivation	Mitochondria	Restored Cell Proliferation and respiration	TNTs	[37]
MSCs	Neural stem cells	Co-culture with cisplatin damaged NSCs	Mitochondria	Normalized mitochondrial membrane potential, prevented cell death	TNTs	[38]
Astrocytes	Neurons	Ischemia	Functional mitochondria	Increased neuronal viability and ATP levels	MVs	[39]
Astrocytes	Neurons	Cisplatin treatment	Mitochondria	Increased neuronal survival, restored neuronal mitochondrial membrane potential, and normalized neuronal calcium dynamics		[40]
Astrocytes or neuronal cells	Astrocytes		Mitochondria	Elevated mitochondrial membrane potential		[41]
hCMEC/D3 (human brain EC cell line)	Brain ECs and neurons	Ischemia	Polarized mitochondria	Increased ATP levels	EVs	[42]
Cardiomyocytes	Cardiofibroblasts		Mitochondria		TNTs	[43]
Cardiac myofibroblasts	Cardiomyocytes	Hypoxia/reoxygenation	Mitochondria	Reduced apoptosis	TNTs	[44]
Human-iPS cell-derived cardiomyocytes (iCMs)	Cardiomyocytes		Mitochondria	Restoration of bioenergetics and mitochondrial biogenesis	EVs	[45]
Renal scattered tubular cells	Tubular epithelial cells		Mitochondria	Improved mitochondrial function (in vitro), perfusion and oxygenation(in vivo)	EVs	[46]
	B16ρ^0^ mouse melanoma cells		Mitochondria	Rescued mitochondrial function		[50]
BMSCs	Multiple myeloma cells		Mitochondria	Increased ATP levels and proliferation	TNTs	[49]
MSCs	Jurkat cells	Chemotherapeutic drugs	Mitochondria	Chemoresistance	TNTs	[51]
MSCs	Primary bone marrow cells of ALL patients	Chemotherapeutic drugs	Mitochondria	Reduced apoptosis	TNTs	[52]
NKT cells (DN32.D3, mouse T cells)	Cancer cells (MDA-MB-231, 4T1, TALL-104)		Mitochondria	Enhanced cancer cell activity and caused death of immune cells	TNTs	[53]
MDA-MB-231 breast cancer cells	MDA-MB-231 breast cancer cells	Activation of mGluR3 by extracellular glutamate	mtDNA	Promote endosomal trafficking and invasiveness	MVs	[63]
Human umbilical cord derived MSC-CM	SH-SY5Y cells	OA(okadaic acid) treatment	Mitochondria	Alleviated oxidative stress, suppressed apoptosis, improved mitochondrial function	MVs	[64]
MSCs	Macrophage	Acute respiratory distress syndrome (ARDS)	Mitochondria	Promote an anti-inflammatory and highly phagocytic macrophage phenotype	MVs	[58]
Macrophages	Sensory neurons	Inflammation	Mitochondria	Resolution of inflammatory pain	EVs	[65]
Airway myeloid-derived regulatorycells	T cells		Polarized mitochondria and mtDNA		EVs	[56]
Neural stem cell	mtDNA-deficient L929 Rho^0^ cells (mononuclear phagocytes)		Functional mitochondria	Rescued mitochondrial function, increased cell survival, reduced the expression of pro-inflammatory markers	EVs	[66]
Platelets	MSC		Mitochondria	Improved the regenerative capacity	EVs	[67]
MSCs	Epithelial cell	Stress induction by rotenone or TNF-α	Mitochondria	Reduced ROS production	TNTs	[68]
MSCs	cardiomyocytes	Anthracycline or Dox	Mitochondria	Rescued damage	TNTs	[69]

NSCLC: Non-small cell lung cancer, EC: endothelial cell, iPSC: induced pluripotent stem cell, BMSC: Bone marrow stromal cells, EV: extracellular vesicles, TNT: tunneling nanotubes.

## Data Availability

Not applicable.

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
