# Peer review of "The Pathophysiological Significance of “Mitochondrial Ejection” from Cells"

_biomolecules, 2022, doi:10.3390/biom12121770_

Round 1

Reviewer 1 Report

In this manuscript, Fan et al provide a comprehensive overview of current knowledge on the intercellular mitochondrial transfer mechanisms.

The manuscript is well-written and provides sufficient details and updates.

I have only minor comments:

1. Line 36: exclude the phrase “for surviving cells via oxidative…”. For many cells, glycolysis is sufficient to survive

2. Line 42-44: Please rewrite this sentence, e.g., “As a result of mitochondrial damage, mtDNA can be released playing detrimental role…”

3. Line 57: Include a reference for the sentence “In addition,…”

4. Line 75: Include a reference for the sentence: “The overexpression of …”

5. Line 78: Include a reference for the sentence: “On the other hand…”

6. Line 80: please explain “acidotic condition”, i.e., pH

7. Line 82: “as large vesicles”. Probably the authors mean “in large vesicles”

8. Lines 95 and 101-102: References are required

9. Line 112: “thermogenic stress”, the authors probably mean a widely used expression “cold stress”. Please change

10. Line 115: Reference is required for the sentence “The in vivo…”

11. Line 119: Explain the expression “mitochondrial particles”

12. Line 119: the statement “...were transferred from stressed adipocyte to CMs...” is wrong. In fact, the isolated cardiomyocytes in vitro took sEVs isolated from adipocytes (Figure 1L). Please correct.

In vivo data (Figure 5D) are not convincing

13. Line 175: Statement: “transfer between CMs…” please note “neonatal CMs”

14: Line 264: Please introduce in text S100A4

15. Lines 271 and 280: again “neonatal CMs”

Author Response

I thank the reviewer for the thorough analysis of the manuscript and valuable comments. I hope that the revised version will now be considered suitable for publication in Biomolecules.

Reviewer 1:

  1. Line 36: exclude the phrase “for surviving cells via oxidative…”. For many cells, glycolysis is sufficient to survive

--- Excluded.

  1. Line 42-44: Please rewrite this sentence, e.g., “As a result of mitochondrial damage, mtDNA can be released playing detrimental role…”

--- Corrected.

  1. Line 57: Include a reference for the sentence “In addition,…”

--- Added a suitable reference [Ref. 24].

  1. Line 75: Include a reference for the sentence: “The overexpression of …”

--- Added a suitable reference [Ref. 15].

  1. Line 78: Include a reference for the sentence: “On the other hand…”

--- Added a suitable reference [Ref. 15].

  1. Line 80: please explain “acidotic condition”, i.e., pH

--- Added an explanation as follows: “the cells were incubated with pH6.5 solution”.

  1. Line 82: “as large vesicles”. Probably the authors mean “in large vesicles”

--- Corrected.

  1. Lines 95 and 101-102: References are required

--- Added a suitable reference [Ref. 22].

  1. Line 112: “thermogenic stress”, the authors probably mean a widely used expression “cold stress”. Please change

--- Corrected.

  1. Line 115: Reference is required for the sentence “The in vivo…”

--- Added a suitable reference [Ref. 16].

  1. Line 119: Explain the expression “mitochondrial particles”

--- Changed the expression “mitochondrial particles” to “EVs containing functional but damaged mitochondria”.

  1. Line 119: the statement “...were transferred from stressed adipocyte to CMs...” is wrong. In fact, the isolated cardiomyocytes in vitro took sEVs isolated from adipocytes (Figure 1). Please correct. 

--- Corrected as follows: “which were produced by stressed adipocytes, were found to enter isolated CMs,”.

  1. Line 175: Statement: “transfer between CMs…” please note “neonatal CMs”

--- Added.

  1. Line 264: Please introduce in text S100A4

--- Added the full name of S100A4 as follows: S100 calcium binding protein A4.

  1. Lines 271 and 280: again “neonatal CMs”

--- Added.

Reviewer 2 Report

The review article from Qintao et al. gives an overview on the current knowledge of the functional significance of mitochondrial transmigration across the plasma membranes in various cells with particular attention to the usefulness of such phenomena to develop novel therapeutic strategies relevant to various diseases, including cardiovascolar disease, providing insights for future studies.

The review is interesting and well argued with proper references although some parts are unclear and should be re-written: for instance, page 1, line 33 to line 35, and page 4, line 139 to line 146. Please, in the latter paragraph include the proper references.

To improve readability I suggest the manuscript to be revised by a native speaker.

I think this review article is a valuable contribution and therefore it should be considered  for publication in Biomolecules.

Author Response

I thank the reviewer for the thorough analysis of the manuscript and valuable comments. I hope that the revised version will now be considered suitable for publication in Biomolecules.

Reviewer2:

  1. The reviewis interesting and well argued with proper references although some parts are unclear and should be re-written: for instance, page 1, line 33 to line 35, and page 4, line 139 to line 146. Please, in the latter paragraph include the proper references.

--- Regarding page 1, line 33 to line 35, we changed the description from “As the majority of genes encoding mitochondrial proteins are transferred...” to “Most of the genes that encode mitochondrial proteins have been transferred...”.

Regarding page 4, line 139 to line 146, we also changed the description from “Namely, the reduction of dynein binding, the inward motor, and the promotion of kinesin family member 5B (KIF5B) binding—the outward motor that belongs to kinesin superfamily—in cooperation with mitochondrial fission regulator dynamin-related protein 1 (Drp1) and myosin XIX (Myo19)-enhanced migration of damaged mitochondria to the peripheral area of cells, which resulted in the disposal of migrasome-mediated damaged mitochondria” to “Damaged mitochondria have a weakened ability to rescue dynein, the inward motor, while their ability to bind kinesin family member 5B (KIF5B) significantly increased. Therefore, in cooperation with mitochondrial fission regulator dynamin-related protein 1 (Drp1) and myosin XIX (Myo19), damaged mitochondria are transferred to the peripheral area of cells and then are disposed of by migrasomes with adding a suitable reference [Ref. 25].

  1. To improve readability I suggest the manuscript to be revised by a native speaker.

--- We asked Mr. Brian Quinn, a native English speaker and a specialist of editing and proofreading services in scientific manuscripts, for editing English of our manuscript (Please refer a certification of English editing).
